Scaredy-cats don’t succeed: behavioral traits predict problem-solving success in captive felidae

O’Connor Victoria L.
Vonk Jennifer vonk@oakland.edu
Psychology, Oakland University , Rochester, MI , United States of America
Drobniak Szymon
Electronic publication date: 2022 Nov 25
Publication date: 2022
Volume: 10
Electronic Location ID: e14453
Received 2022 May 5; Accepted 2022 Nov 2
Copyright: © 2022 O’Connor and Vonk
Copyright year: 2022
Copyright holder: O’Connor and Vonk
License: This is an open access article distributed under the terms of the Creative Commons Attribution License, which permits unrestricted use, distribution, reproduction and adaptation in any medium and for any purpose provided that it is properly attributed. For attribution, the original author(s), title, publication source (PeerJ) and either DOI or URL of the article must be cited.
License URL: https://creativecommons.org/licenses/by/4.0/

Keywords: Cats, Individual differences, Keeper assessment, Innovation, Behavioral flexibility 

Funding: The authors received no funding for this work.

==============================
Behavioral traits can be determined from the consistency in an animal’s behaviors across time and situations. These behavioral traits may have been differentially selected in closely related species. Studying the structure of these traits across species within an order can inform a better understanding of the selection pressures under which behavior evolves. These adaptive traits are still expected to vary within individuals and might predict general cognitive capacities that facilitate survival, such as behavioral flexibility. We derived five facets (Flexible/Friendly, Fearful/Aggressive, Uninterested, Social/Playful, and Cautious) from behavioral trait assessments based on zookeeper surveys in 52 Felidae individuals representing thirteen species. We analyzed whether age, sex, species, and these facets predicted success in a multi access puzzle box–a measure of innovation. We found that Fearful/Aggressive and Cautious facets were negatively associated with success. This research provides the first test of the association between behavioral trait facets and innovation in a diverse group of captive felidae. Understanding the connection between behavioral traits and problem-solving can assist in ensuring the protection of diverse species in their natural habitats and ethical treatment in captivity.

Introduction

Personality generally refers to consistency in behavioral traits across time and situations. Applying the term to nonhuman animals has sometimes been considered controversial because of its anthropomorphic nature (Gosling & Vazire, 2002). Therefore, researchers often refer to behavioral traits, rather than personality, when working with nonhumans, and we will follow that convention. Behavioral traits will be used here to refer to individual differences in the expression of behavioral tendencies such as decision-making, risk taking, subjective wellbeing, and coping strategies (Dall, Houston & McNamara, 2004) that are consistent across time and context. We explore the association between these traits and the ability of individuals of various felid species to solve a task designed to measure innovation.

Behavioral traits are influenced by natural selection through genetic and/or environmental effects (Moore, Brodie & Wolf, 2017). Studies have found that, in more than 100 species ranging from insects to mammals, conspecifics, independent from sex or age, differ profoundly in their behavior (e.g., Carere, Caramaschi & Fawcett, 2010; Dougherty & Guillette, 2018). Additional studies of behavioral traits have shown varied effects on health and longevity, including immune function (e.g., Capitanio et al., 2008), morbidity (e.g., Natoli et al., 2005), chronic stress (e.g., Wielebnowski et al., 2002), and mortality (e.g., Weiss et al., 2013). In addition to heritability and fitness (e.g., Sinn, Apiolaza & Moltschaniwskyj, 2006), social foraging and collective behavior (e.g., Aplin et al., 2014), swarm intelligence for defense, resource exploitation and hunting (e.g., Krause, Ruxton & Krause, 2010), well-being (e.g., King & Landau, 2003; Weiss et al., 2009; Weiss, King & Perkins, 2006), and predator avoidance (e.g., Handegard et al., 2012) have also been linked to these individual differences. Personality is relevant for improving zoo management (e.g., individual differences suggest some may require more seclusion in enclosures; Wielebnowski, 1999), animal welfare (e.g., females score higher as tense-fearful than males; Wielebnowski, 1999), captive breeding (e.g., tense-fearful cheetahs are less likely to reproduce in captivity; Wielebnowski, 1999), enclosure grouping (e.g., age, number of individuals, and sex ratio are important for social housing; Stoinski et al., 2004), and conservation (e.g., bolder, less fearful individuals were less suitable for wild release; Bremner-Harrison, Prodohl & Elwood, 2004). Therefore, knowledge of an individual’s traits can predict how that individual may perform in a novel situation or task.

In captivity, individual behavioral traits affect animals’ experiences and predict their adjustment and behaviors. Animal behavioral traits are primarily assessed in three ways: keeper assessments, behavioral coding, and preference tests (Watters & Powell, 2012). Keeper assessments involve familiar individuals rating the predefined traits of each subject on a scale based on their human-animal relationships (HAR), knowledge of the subject accumulated across time, mutual recognition between the human and animal, and the nature of interactions (positive or negative). Good HARs require the human and animal to have a history of positive interactions to allow successful behavioral predictions based on the caregiver’s experience witnessing an animal’s consistent likelihood to engage in different behaviors during the length of their relationship (Estep, 1992). In this manner, survey assessments also capture consistency in behavior across time and situations, but these assessments are subjective. First, raters can interpret trait definitions differently. For example, raters may define “flexible” differently based on their own experiences and biases. Whereas in the survey used here, flexible is defined as “adapts comfortably to change,” change can be defined as a shift in exhibits, conspecifics, and/or zookeepers. Keepers may use different experiences as reference points for their ratings based on their own interactions with the animal. Similarly, one keeper may define “bold” as reacting aggressively to novelty, while another may see that as acting fearfully and score the individual low on boldness. In addition, keepers may allow their own biases to color their ratings; for example, they may be more likely to rate male animals high and to rate female animals low on boldness even when the animals behave similarly. Thus, the use of keeper assessments is strengthened when multiple keepers have built a relationship with the subject and reliability can be determined.

Another potential issue with keeper assessments is that the traits animals are rated for are often derived from top-down models that may not fit the target species. Thus, it is important to first understand the traits that best describe the variability in the behavior of members of the study species (Vonk & Eaton, 2018). Of the various possible forms of keeper assessment identified by Uher (2011), we adopted a lexical top-down approach, using behavioral trait descriptors from closely related species to fit the current subjects. Top-down assessments begin with an existing model of personality and attempt to assess the extent to which a given species exhibits traits derived from that model. These assessments seek to identify facets—sets of definable traits that correlate and can be grouped together under an umbrella term, akin to behavioral syndromes (Sih et al., 2004). Early research involving nonhuman animals focused on the popular five-factor model derived from humans (McCrae & Costa, 1987; Tupes & Christal, 1992), which includes the facets of openness, conscientiousness, neuroticism, extraversion, and agreeableness.

Factor analytic approaches have revealed that the structure of behavioral syndromes in nonhumans may differ from that established in humans. For example, in primates, behavioral syndromes are described by the combination of two or more of the following six facets: dominance, extraversion, dependability, emotional stability, agreeableness, and openness (Weiss, King & Figueredo, 2000). In canids, there is less agreement regarding which key traits compose the structure of canid behavioral syndromes. Domestic dogs exhibit a variety of traits that include extraversion, neuroticism, agreeableness, openness/conscientiousness (Gosling & John, 1999), playfulness, curiosity/fearlessness, chase-proneness, sociability, and aggressiveness (Svartberg & Forkman, 2002).

Of particular relevance to the current study, Stanton, Sullivan & Fazio (2015) conducted an in-depth meta-analysis on the current Felidae literature, ultimately creating a standardized Felidae ethogram. Ethograms include the documentation of species-exhibited behaviors by knowledgeable individuals (Stanton, Sullivan & Fazio, 2015) and allow behavioral tracking of captive populations that can be informative for predicting reproduction and overall welfare (Clubb & Mason, 2003). Most relevant studies on felids have derived models consisting of the following six facets: active, aggressive, curious, dominant, sociable, and timid/fearful/tense (Gartner & Weiss, 2013). For instance, in cheetahs (Acinonyx jubatus), Wielebnowski (1999) documented three major behavioral facets—tense-fearful, excitable-vocal, and aggressive whereas Phillips et al. (2017) identified three facets—nervousness, adventurousness, and aggression using keeper observations. Thus, similar facets emerged in both samples of cheetahs from studies conducted 18 years apart. Gartner, Powell & Weiss (2014) measured behavioral traits in five felid species. Keepers were asked to rate the cats on the same behavioral traits for each species. Although slightly different facets emerged from a factor analysis including neuroticism, dominance, and impulsiveness in African lions, neuroticism, agreeableness/openness, and dominance/impulsiveness in clouded leopards, neuroticism, impulsiveness/openness, and dominance in snow leopards, dominance, impulsiveness, and neuroticism in domestic cats, and dominance, agreeableness, and self-control in Scottish wildcats, there were common facets that appeared to characterize Felidae in general (Gartner, Powell & Weiss, 2014). This study lays the foundation for the current research, which examines variability within and between felid species. The current research drew upon several previously used keeper assessments (Carlstead, Mellen & Kleiman, 1999; Gartner & Weiss, 2013; Gold & Maple, 1994; Martin-Wintle et al., 2017; Phillips & Peck, 2007; Wielebnowski, 1999; Wielebnowski et al., 2002) to incorporate the traits: active, anxious, calm, cautious, cooperative, curious, dominant, excitable, fearful, flexible, playful, smart, sociable, solitary, stereotypical, submissive, tense, vigilant, and uninterested.

Having established the structure of behavioral traits in a taxa like felids, one can then examine whether the derived traits usefully predict behaviors in various contexts. Testing contexts, as defined by Freeman, Gosling & Schapiro (2011), incorporate the subjects’ responses to a novel stimulus to elicit differing reactions from the subjects to document individual differences. The initial studies of Carlstead, Mellen & Kleiman (1999) and Powell & Svoke (2008) introduced the idea that observations of subjects’ interactions with novel enrichment provides insight into their behavioral traits. Carlstead, Mellen & Kleiman (1999) paired keeper assessments with a novel object test and a novel conspecific scent test. Using a 52-trait and behavior assessment that they developed, keepers from different zoos were able to reliably differentiate black rhinoceros individuals (Diceros bicornis) based on their physical characteristics (e.g., sex, origin, and age) and six behavioral traits (e.g., olfactory behaviors, chasing/stereotypy/mouthing, fear, friendly to keeper, dominant, and patrolling). Similarly, Powell & Svoke (2008) evaluated giant pandas’ (Ailuroproda melanoleuca) responses to ten novel enrichment items and their 23 trait and behavior assessment using keeper responses, allowing them to create individual behavioral profiles.

Studies have demonstrated that the novel—object test can be a reliable and valid behavioral trait tool in Felidae. Gartner & Powell (2012) used keeper assessments and coded behaviors in response to six novel objects to identify five dimensions—active/vigilant, curious/playful, calm/self-assured, timid/anxious, and friendly to humans—differentiating snow leopards (Panthera uncia) based on age and sex. Similarly, Phillips et al. (2017) examined four behavioral trait states in tigers (Panthera tigris), including aggression, fear, vigilance, and obedience; this time, using both keeper assessments and behaviors towards olfactory and physical enrichment. Ratings from behavioral trait assessments correlate with performance on novel object tests validating the use of behavioral trait ratings.

The current work extends the existing literature demonstrating that behaviors elicited by novel tasks are useful in validating zookeeper assessments of captive carnivore behavioral traits by assessing whether keeper assessments can predict performance on a novel problem-solving task for environmental and cognitive enrichment. Here, the multi-access puzzle box (MAB) as described in O’Connor et al. (2022) is used as a test of innovation in felids. Various authors (Benson-Amram, Weldele & Holekamp, 2013; Benson-Amram et al., 2016; Daniels et al., 2019; Johnson-Ulrich, Johnson-Ulrich & Holekamp, 2018; O’Connor et al., 2022), found that behavioral measures of high persistence, high motor diversity/exploration diversity, high activity/working time, and low neophobia are associated with success on a MAB in carnivores. Behavioral trait facets similar to these behaviors are expected to predict performance in a task designed to measure behavioral flexibility. For example, traits such as ‘Cautious’ and ‘Anxious’ might relate to neophobia, whereas ‘Playful’ and ‘Curious’ might relate to exploration. There have been numerous efforts to relate behavioral traits to individual differences in cognition. For example, the expression of particular traits can influence cognitive abilities such as success or failure (Carere & Locurto, 2011). However, there is, as of yet, no clear unifying theory about the expected association between personality and cognition in nonhumans (Griffin, Guillette & Healy, 2015; Sih & Giudice, 2012).

Individual differences in behavioral traits are important for determining the best fit practices for captive husbandry (e.g., Goswami et al., 2020), well-being (e.g., Gartner, Powell & Weiss, 2016), enrichment preference (e.g., Wang et al., 2019), health and reproduction (e.g., Wielebnowski, 1999), social compatibility (e.g., Bullock, James & Williams, 2021), social group dynamic roles (e.g., Dunston et al., 2016), and environmental/management changes (e.g., Pastorino et al., 2017). Additionally, activity/stress levels (e.g., Torgerson-White & Bennett, 2014) have been shown to predict behavioral responses across a variety of taxa, including carnivores. The current research extracts behavioral trait facets from keeper assessments to explore whether these facets predict success on a MAB box, which measures innovation, in 52 individuals representing 13 species of felids.

Materials and Methods

Species and rater information

Subjects included 52 individuals, 30 males and 22 females, from 13 species (see Table 1). The age of the subjects ranged from 6 months to 23-years-old (M = 6.68, SD = 5.96). Raters include thirty-seven keepers who spent, on average, 2.2 years with subjects (SD = 2.19) from five locations: the Bergen County Zoo (BCZ) in Paramus, New Jersey, the Bronx Zoo (BZ) in Bronx, New York, The Creature Conservancy (TCC) in Ann Arbor, Michigan, the Oklahoma Zoo (OKC) in Oklahoma City, Oklahoma, and the Turtle Back Zoo (TBZ) in West Orange, New Jersey. Because keepers and subjects were housed at different institutions, the same keepers did not rate all subjects.

Table 1 Descriptive information for all subjects.

Name	Species	Sex	Age (yrs)	Rearing	Zoo	
Amara	African lion (Panthera leo)	F	5	Captive	TBZ	
Bahati	African lion (Panthera leo)	M	5	Captive	BZ	
Demarcus	African lion (Panthera leo)	M	4	Captive	TBZ	
Huey	African lion (Panthera leo)	M	10	Captive	OKC	
Ime	African lion (Panthera leo)	M	5	Captive	BZ	
Sukari	African lion (Panthera leo)	F	15	Captive	TBZ	
Thulani	African lion (Panthera leo)	M	5	Captive	BZ	
Annika	Amur leopard (Panthera pardus orientalis)	F	5	Captive	TBZ	
Nadya	Amur leopard (Panthera pardus orientalis)	F	½	Captive	TBZ	
Valeri	Amur leopard (Panthera pardus orientalis)	M	7	Captive	TBZ	
Astrid	Bobcat (Lynx rufus)	F	23	Captive	TBZ	
Dodger	Bobcat (Lynx rufus)	M	2	Wild	OKC	
ZZ	Caracal (Caracal caracal)	F	18	Captive	OKC	
Alvin	Cheetah (Acinonyx jubatus)	M	1	Captive	TBZ	
Nandi	Cheetah (Acinonyx jubatus)	F	1	Captive	TBZ	
Simon	Cheetah (Acinonyx jubatus)	M	1	Captive	TBZ	
Theodore	Cheetah (Acinonyx jubatus)	M	1	Captive	TBZ	
JD	Clouded leopard (Neofelis nebulosa)	M	3	Captive	OKC	
Jye	Clouded leopard (Neofelis nebulosa)	M	1	Captive	TBZ	
Madee	Clouded leopard (Neofelis nebulosa)	F	½	Captive	TBZ	
Mali	Clouded leopard (Neofelis nebulosa)	F	1	Captive	TBZ	
Rukai	Clouded leopard (Neofelis nebulosa)	F	3	Captive	OKC	
Chinook	Cougar (Puma concolor)	M	4	Wild	BCZ	
Harper	Cougar (Puma concolor)	F	5	Wild	TCC	
Jane	Cougar (Puma concolor)	F	1	Wild	TBZ	
Josey	Cougar (Puma concolor)	F	1	Wild	TBZ	
Sage	Cougar (Puma concolor)	F	15	Captive	TBZ	
Tacoma	Cougar (Puma concolor)	M	4	Wild	BCZ	
Wyatt	Cougar (Puma concolor)	M	1	Wild	TBZ	
Boon	Fishing cat (Prionailurus viverrinus)	M	7	Captive	OKC	
Chet	Fishing cat (Prionailurus viverrinus)	M	12	Captive	OKC	
Miri	Fishing cat (Prionailurus viverrinus)	F	15	Captive	OKC	
Puddles	Fishing cat (Prionailurus viverrinus)	M	4	Captive	OKC	
Rosa	Jaguar (Panthera onca)	F	9	Captive	TBZ	
Tai	Jaguar (Panthera onca)	M	17	Captive	OKC	
Arieta	Ocelot (Leopardus pardalis)	F	8	Captive	OKC	
Bosco	Ocelot (Leopardus pardalis)	M	13	Captive	OKC	
Old Man	Ocelot (Leopardus pardalis)	M	20	Captive	BCZ	
Makusi	Ocelot (Leopardus pardalis)	M	1	Captive	BCZ	
Raif	Ocelot (Leopardus pardalis)	M	8	Captive	OKC	
Nanai	Siberian lynx (Lynx lynx wrangeli)	M	½	Wild	TCC	
Chameli	Snow leopard (Panthera uncia)	F	6	Captive	TBZ	
Gala	Snow leopard (Panthera uncia)	M	4	Captive	TBZ	
K2	Snow leopard (Panthera uncia)	F	8	Captive	BZ	
Khyber	Snow leopard (Panthera uncia)	F	1	Captive	BZ	
Leo	Snow leopard (Panthera uncia)	M	13	Wild	BZ	
Mike	Snow leopard (Panthera uncia)	M	4	Captive	BZ	
MJ	Snow leopard (Panthera uncia)	M	2	Captive	BZ	
Tanja	Snow leopard (Panthera uncia)	F	18	Captive	BZ	
Willie	Snow leopard (Panthera uncia)	M	5	Captive	BZ	
Kami	Sumatran tiger (Panthera tigris sumatrae)	M	14	Captive	OKC	
Lola	Sumatran tiger (Panthera tigris sumatrae)	F	10	Captive	OKC	

Testing was approved by the IACUCs at Oakland University (# 19111), The City University of New York: Hunter College (#SC-Captive 4/21), and The Wildlife Conservation Society (#18:01).

Carnivore behavior survey and procedure

To properly compare the ability of behavioral traits to predict success in the MAB across all felids, individuals were not assigned species-unique traits. Instead, individuals were assessed on the same traits to determine whether individual or species level variation better predicts success in this task. Gosling & John (1999) reviewed 19 factor analytic studies across 12 nonhuman species verifying considerable generality in non-related species. In an effort to examine behavioral differences in captive Felidae, Stanton, Sullivan & Fazio (2015) analyzed surveys of 30 species and 40 subfamilies to find that most behaviors identified in their study were similarly described and likely to apply to most felid species. In a similar review of 20 published studies, Gartner & Weiss (2013) found reasonable consistency of certain personality dimensions in felids. As such, we expected that the behaviors of the 13 species studied here would have similarly structured behavioral traits, Furthermore, all felids share morphological features such as binocular vision, flexible bodies with muscular limbs, protracting claws, and external ears with similar levels of hearing (Kitchener et al., 2010; Rothwell, 2003). However, it is important to note that it was not the goal of the current study to determine or compare the structure of behavioral syndromes in individual Felidae species.

The twenty-seven-item behavioral trait survey was developed based on previous surveys (Feaver, Mendl & Bateson, 1986; Gartner & Powell, 2012; Stanton, Sullivan & Fazio, 2015; Wielebnowski, 1999), and included the trait “intelligent.” However, we did not include scores on “intelligent” in our analyses as it is not clear that this should be considered a personality, rather than a cognitive trait, and, as such, the results predicting success on a problem-solving task could be circular. Each item in the survey included a specific description. For example, Active, was described as “moves about a lot” (see Table 2). Four traits, Aggressive, Fearful, Friendly, and Uninterested, were rated with regard to three contexts-overall, with novelties or environmental changes, and with humans. All traits were rated on an eight-point Likert scale, where 0 = Doesn’t apply, 1 = Does not describe at all, 4 = Neutral, and 7 = Describes very well.

Table 2 Definitions of traits used in the keeper assessment survey.

Adjective	Definition	
Active	Moves about a lot	
Anxious	Uneasy, easily startled	
Calm	Not easily disturbed by changes within or outside environment	
Cautious	Exhibits care in actions	
Cooperative	Easily compliant	
Curious	Readily explores new situations	
Dominant	Displaces/overpowers others	
Excitable	Strong reaction to changes	
Fearful	Easily shaken; avoids changes and assumes protective or aggressive body postures	
Flexible	Adapts comfortably to change	
Playful	Initiates and easily joins in play	
Sociable	Seeks out companionship	
Solitary	Chooses to spend time alone	
Stereotypical	Fixed and oversimplified in behaviors	
Submissive	Gives in easily to others	
Tense	Shows restraint in posture and movement; carries the body stiffly and tries to pull back and be less noticeable	
Vigilant	Alert, attentive, notices all changes	
Uninterested	No care in changes in environment, or conspecifics	
With novelties or environmental changes	
Aggressive	Hostile or threatening reaction	
Fearful	Retreats from others	
Friendly	Initiates proximity	
Uninterested	Shows no interest	
With humans	
Aggressive	Hostile or threatening rection	
Fearful	Retreats from people	
Friendly	Initiates proximity	
Uninterested	Shows no interest	

Each keeper was given the questionnaire individually and instructed not to consult others, so that their responses reflected their independent ratings of the individual subjects. Keepers were asked to provide the following information about themselves: age, sex, and years of experience with big cats, the species, the individual, and their zoo. In most cases, keepers completed the questionnaires without knowledge of how individuals performed in the MAB although this was not the case for a subset of the cats tested at OKC (n = 10). Given their long-term experience with the individuals, zookeepers’ ratings of felid traits are unlikely to have been impacted by the individual’s performance in the small number of experimental trials in a single task that were observed by any given keeper. Furthermore, any bias resulting from witnessing test sessions was expected to be limited to intelligence and we opted not to include intelligence in our model. Therefore, to retain the largest and most inclusive sample, we opted not to remove data from these subjects.

Problem-solving task and procedure

Upon completion of the surveys by the keepers, a problem-solving task, which involved retrieving a food reward from a custom multi-access puzzle box (MAB) was presented. This task presents a simple and effective behavioral test for exploring innovation and has been used successfully in a variety of carnivores (O’Connor et al., 2022).

All subjects were tested individually in their indoor, or outdoor, off-exhibit holding enclosures. The custom multi-access puzzle boxes were two molded Starboard boxes with stainless-steel frames measuring 0.6 m × 0.6 m × 0.6 m and 0.38 m × 0.38 m × 0.38 m. A food reward placed inside the box was accessible via three separate solutions: (1) Push Door Technique (see Fig. 1); (2) Pull Rope Technique (see Fig. 2); and (3) Pull Door Technique (see Fig. 3). Each solution was presented on a different side of the box. The puzzle box was cleaned and disinfected between different species’ trials and subjects could not see each other during trials.

Figure 1 The multi-access puzzle box showing the open push door technique, or solution one which is opened by pushing the door allowing for access to the food reward.

Figure 2 The multi-access puzzle box showing the pull rope technique, or solution two which swings open by pulling the rope exposing the inside of the box.

Figure 3 The multi-access puzzle box showing the open pull door technique, or solution three which pulls down flush to the ground, exposing the entire inside of the box.

Subjects underwent one trial per day. The trial began when the subject made physical contact with the puzzle box. Trials ended when the subject opened the puzzle box (a successful trial) or after 15 min elapsed without the subject opening the puzzle box (a failed trial). At the end of each trial, the subject was shifted to an adjacent enclosure according to the zoos’ procedures. A subject either failed a condition, which was defined as failing to open the box in three out of five trials, or succeeded in a condition, which was defined as opening the box in three out of five trials. Subjects that succeeded moved on to the next condition. Subjects that failed did not advance to the next condition and testing was discontinued.

Condition 1 (five trials): The reward was retrievable via any of the solutions; all three doors were unlocked at the start of the first trial. Once a subject achieved their first successful trial, the door that they opened remained unlocked and the other two doors were locked for the remainder of the first condition. Three successful trials out of a possible five advanced the subject to the next condition. Condition 2 (five trials): The remaining two unsolved doors were unlocked at the start of the first trial. Once a subject succeeded in opening an unlocked door, that door remained unlocked and the other two doors were locked for the remainder of the second condition. Three successful trials out of a possible five advanced the subject to the final condition. Condition 3 (five trials): Only the remaining unsolved door was unlocked, and the subject was given five trials in which to open it three times, ending testing.

Statistical analysis

Data were analyzed using SPSS v. 28 software for Macintosh. Results were considered significant at alpha level p < 0.05.

For individuals that had more than one keeper rating their behavioral trait, interrater reliabilities were calculated using the Intraclass Correlation Coefficients (ICC) where we examined consistency between multiple raters using the model for Case 1 described by Shrout & Fleiss (1979) where each subject was rated by a different set of randomly selected raters that did not rate all subjects. Subjects of the same species at the same facility may have been rated by the same raters but the raters differed by species and facility. Items deemed unreliable, defined as having an ICC of less than or equal to zero, were omitted from further analysis. The average of the keepers’ ratings was then used when there were multiple raters for a subject.

Parallel analysis (Horn, 1965; O’Connor, 2000) was used as additional confirmation of the number of facets to be extracted from the survey data and both the mean and percentile estimations of model fit supported a five-component model to best fit the data. Subsequently, a principal components analysis (PCA) was conducted using a varimax rotation and five factors were extracted to combine the reliable behavioral traits into behavioral facets. Traits were assigned to the facet where they had the highest loading. Traits with negative loadings were reverse scored and composite variables taking the average ratings for each loaded trait were created. Only loadings of 0.40 or greater were considered. Fig. 4 depicts the structure of the first three facets.

Figure 4 Structure of the first three facets extracted from the PCA.

In their interactions with the MAB, individual carnivores were coded on their Success (0 = no solutions opened, 1 = success on at least one condition). At least two independent observers verified the classification of trials as successful. Additional data from this task will be reported elsewhere. We regressed success on to sex, age, and subfamilies (1 = Pantherinae, 2 = Felinae) in the first step of a hierarchical logistic regression model using the Wald Chi-square test to determine whether any of the predictors significantly contributed to the outcome. We entered the five behavioral trait facets derived from the PCA in the second step of the model. Independent samples t-tests were conducted to determine whether the subfamilies differed on any of the five facets.

Results

Keeper assessment interrater reliability

The reliabilities across raters, ICC were 0.37 (active), 0.41 (anxious), 0.39 (calm), 0.12 (cautious), 0.07 (cooperative), 0.44 (curious), 0.57 (dominant), 0.33 (excitable), 0.54 (fearful), 0.41 (flexible), 0.51 (playful), 0.23 (smart), 0.60 (sociable), 0.50 (solitary), 0.11 (stereotypical), 0.67 (submissive), 0.23 (tense), 0.05 (vigilant), 0.62 (uninterested), 0.18 (aggressive to novelty), 0.05 (fearful of novelty), 0.36 (friendly towards novelty), 0.63 (uninterested in novelty), 0.28 (aggressive to humans), 0.38 (fearful of humans), 0.14 (friendly to humans), and 0.36 (uninterested in humans). The reliabilities of mean ratings, ICC were 0.75 (active), 0.77 (anxious), 0.76 (calm), 0.40 (cautious), 0.26 (cooperative), 0.80 (curious), 0.87 (dominant), 0.71 (excitable), 0.85 (fearful), 0.78 (flexible), 0.84 (playful), 0.60 (smart), 0.88 (sociable), 0.83 (solitary), 0.39 (stereotypical), 0.91 (submissive), 0.60 (tense), 0.22 (vigilant), 0.89 (uninterested), 0.52 (aggressive to novelty), 0.21 (fearful of novelty), 0.74 (friendly towards novelty), 0.89 (uninterested in novelty), 0.66 (aggressive to humans), 0.75 (fearful of humans), 0.45 (friendly to humans), and 0.73 (uninterested in humans).

Reduction to five facets

The parallel analysis and PCA reduced twenty-six of the behavioral traits to five factors with factor loadings ≥0.40. For these five factors, all PCA values were greater than the eigenvalues derived from the parallel analysis, validating their use (Konečná et al., 2012). Upon examination of the extracted factors, we assigned all traits with values 0.40 or greater (e.g., Gartner & Weiss, 2013) to factors for which they had the highest factor loadings. Thus, we created composite facets representing the following five conceptually coherent facets-Flexible/Friendly, Fearful/Aggressive, Social/Playful, Uninterested and Cautious (see Table 3). The facets Flexible/Friendly, Fearful/Aggressive, and Social/Playful aligned well with previous research with felids (Gartner & Weiss, 2013). The facet Cautious might also be seen as aligning well with Neurotic and Nervousness, which were extracted by prior studies (Gartner & Weiss, 2013; Phillips et al., 2017).

Table 3 Component matrix of six facets reduced from twenty-six behavioral traits.

	Flexible/Friendly	Fearful/Aggressive	Uninterested	Social/Playful	Cautious	
Flexible	0.82					
Curious	0.76					
Friendly to Novelty	0.74					
Cooperative	0.70					
Friendly to Humans	0.69					
Calm	0.60					
Aggressive to Novelty		0.71				
Anxious		0.70				
Stereotypical		0.63				
Excitable		0.60				
Fearful		0.60				
Fearful to Novelty		0.60				
Tense		0.58				
Aggressive to Humans		0.58				
Fearful of Humans		0.54				
Uninterested in Novelty			0.86			
Uninterested			0.82			
Uninterested in Humans			0.78			
Vigilant			−0.47			
Sociable				0.79		
Playful				0.68		
Submissive				0.64		
Cautious					0.73	
Dominant					−0.47	
Active					−0.46	

Behavioral traits predict success

Descriptive statistics and zero order correlations among the variables are shown in Table 4. Success was negatively correlated with Fearful/Aggressive (r = −0.22, p < 0.01), and Cautious (r = −0.24, p < 0.01). Sex was negatively correlated with Fearful/Aggressive (r = −0.18, p < 0.05). Age was positively correlated with Flexible/Friendly (r = 0.19, p < 0.05), and negatively correlated with Social/Playful (r = −0.45, p < 0.01) and Cautious (r = −0.20, p < 0.05). Flexible/Friendly was also positively correlated with Social/Playful (r = 0.29, p < 0.01), and negatively correlated with Fearful/Aggressive (r = −0.54, p < 0.01), Uninterested (r = −0.22, p < 0.01), and Cautious (r = −0.29, p < 0.01). Social/Playful was negatively correlated with Fearful/Aggressive (r = −0.23, p < 0.01) and Uninterested (r = −0.19, p < 0.05).

Table 4 Descriptive statistics and correlations among the variables.

	1	2	3	4	5	6	7	8	9	
1. Success	–									
2. Sex	0.10	–								
3. Age	0.05	0.07	–							
4. Subspecies	0.12	−0.11	−0.01	–						
5. Flexible/Friendly	0.12	0.01	0.19*	0.15	–					
6. Fearful/Aggressive	−0.22**	−0.18*	−0.10	0.00	−0.54**	–				
7. Uninterested	−0.06	−0.01	0.07	−0.07	−0.22**	0.13	–			
8. Social/Playful	−0.02	0.01	−0.45**	0.01	0.29**	−0.23**	−0.19*	–		
9. Cautious	−0.24**	0.12	−0.20*	0.03	−0.29**	0.17*	0.11	−0.16	–	
Mean	0.44	1.48	6.3	1.56	4.28	3.06	3.12	3.51	3.86	
Standard Deviation	0.50	0.50	5.89	0.50	1.17	1.17	1.15	1.69	1.13	
Note:

* p < 0.05

** p < 0.01

Sex, age, and subfamilies were entered into the first step of the logistic regression and the five behavioral facets were entered in the second step to predict success. Age (p = 0.63), sex (p = 0.21), and subfamilies (p = 0.13) did not significantly predict success. Fearful/Aggressive (B = −0.37, SE = 0.16, Wald = 5.48, p = 0.02) and Cautious (B = −0.54, SE = 0.17, Wald = 9.67, p < 0.001) significantly negatively predicted success. Individuals that were rated as more fearful or aggressive and individuals that were rated as more cautious were less likely to have success on the MAB. Flexible/Friendly (B = 0.17, SE = 0.15, Wald = 1.26, p = 0.26), Uninterested (B = −0.09, SE = 0.15, Wald = 0.348, p = 0.56), and Social/Playful (B = −0.01, SE = 0.12, Wald = 0.00, p = 0.97) were not significant predictors of success.

The independent samples t-tests comparing subfamilies, Pantherinae and Felinae, for the five behavioral facets revealed no significant differences (all ps > 0.09).

Discussion

Innovation, as a component of behavioral flexibility, is critical for enabling animals to adapt to changing environments. Species and individuals differ in the extent to which they exhibit behavioral flexibility. Identifying behavioral traits that predict flexibility may facilitate captive husbandry strategies and especially conservation efforts. We examined whether captive carnivore behavioral traits predicted innovation, measured as success on a MAB. A twenty-seven-item keeper assessment survey reduced to five behavioral facets-Flexible/Friendly, Fearful/Aggressive, Uninterested, Social/Playful, and Cautious.

Within felids, the most robust behavioral facets from prior research are Sociable, Dominant, and Curious (Gartner & Weiss, 2013). Our PCA analysis with a greater diversity of species identified similar facets-our Social/Playful to their Sociable, our reverse-scored Cautious to their Dominant, and Fearful/Aggressive and Cautious to Curious. This consistency across studies suggests that felids may share similar trait structures. However, our study is important in suggesting that an individual’s traits, rather than species differences, seem to predict success in the task. Evidence in other species supports this conclusion. Black rhinoceros individuals rated as more fearful also exhibited a longer latency to contact a novel object (Carlstead, Mellen & Kleiman, 1999) and rainbow trout (Onchorhyncus mykiss) rated as shy but given emboldening experiences became bolder and exhibited reduced latencies to contact a novel object (Frost et al., 2007). In the current study, individuals that were rated as more fearful or aggressive and individuals that were rated as more cautious were less likely to have success on the MAB. However, it is important to note that we tested success on only a single problem-solving task so future research is needed to test the generalizability of these associations.

The behavioral traits assessed here were rated with reasonable reliability across keepers and predicted performance in the MAB, which measured innovation. Specifically, Fearful/Aggressive and Cautious predicted lack of success on the MAB. Highly fearful and cautious animals should be less likely to attempt solutions to novel problems, so these results were expected. Although most of the keepers completed the assessments with no knowledge of the subjects’ performance, some keepers for ten subjects may have been influenced in their ratings by observing a subset of the experimental trials. Because we did not include the trait “intelligence” in our analyses, which was the trait that should be most influenced by the subject’s performance, and because no keeper observed all trials, and each keeper had extensive experience of the subjects beyond observing them participate in these trials, we are not concerned that this biased our results. Ideally, studies should have all raters complete their ratings before any of the tests are conducted. This was not always possible here based on the complications of arranging testing at multiple locations during a global pandemic.

Previous studies have identified behavioral traits and facets in felids, some in conjunction with a novel object test (Carlstead, Mellen & Kleiman, 1999; Gartner & Powell, 2012; Powell & Svoke, 2008; Razal, Pisacane & Miller, 2016). This is the first study to report an association of behavioral traits with success on a test of innovation. Diverse behaviors have been associated with problem-solving success in carnivores (Benson-Amram, Weldele & Holekamp, 2013; Benson-Amram et al., 2016; Daniels et al., 2019; Johnson-Ulrich, Johnson-Ulrich & Holekamp, 2018; O’Connor et al., 2022). Based on these findings, we expected Flexible/Friendly and Social/Playful to be predictive of success; however, this was not the case. We defined Flexible as “adapts comfortably to change,” Friendly as “initiates proximity,” and Curious as “readily explores new situations.” Keeper ratings should be informed by knowledge of the animal’s behavior in multiple contexts (e.g., shifting indoor and outdoor enclosures, proximity to zoo guests, etc.). Similarly, we defined Sociable as “seeks out companionship” and Playful as “initiates and easily joins in play.” However, these terms may be related to age as younger cats were rated as more Social/Playful (r = −0.45, p < 0.001), or to socially housed species, of which some of these subjects are not. It is possible that performance in a single problem-solving task is not a representative measure of the subject’s more general abilities. Because neophobia may limit interaction with the novel puzzle box (e.g., coyotes; Young, Touzot & Brummer, 2019), the facets of fearful and cautious may have overshadowed other behavioral traits in predicting success. These results support previous findings suggesting that motivation and object exploration may be better predictors of success compared to cognitive ability or inhibitory skills (Johnson-Ulrich, Johnson-Ulrich & Holekamp, 2018). This point is important given how frequently cognitive research is conducted to compare cognitive abilities between species (Vonk et al., 2021).

There are other nonsignificant results to note. Our research does not corroborate previous findings that age (e.g., Benson-Amram & Holekamp, 2012) or sex (e.g., Amici et al., 2019) predicted problem-solving. It is important to note that our measure of success was only a very cursory measure of performance in this task. The MAB allows for examination of multiple measures of cognition (e.g., trials to success, number of successful trials, number of solutions learned, latency to learn new solution) and behavior (e.g., number of behaviors performed, perseveration), but we examined only the simplest outcome here as a pilot test of how well behavioral traits could predict problem-solving success, which might be associated with adaptability and flexibility to change in novel environments. Thus, we would encourage future researchers to examine how individual differences predict variable success in tasks that might assess traits relevant for species’ survival in the wild or ability to adapt in captivity.

Conclusions

Many studies of animal cognition have a low sample size, (Shaw & Schmelz, 2017). To our knowledge, this study includes data from the largest and most inclusive sample of felids compared to previous studies of felid personality and cognition. Across the thirteen Felidae species we assessed, a coherent behavioral trait structure was extracted involving five facets-Flexible/Friendly, Fearful/Aggressive, Uninterested, Social/Playful, and Cautious. These facets are echoed in the Felidae literature as reviewed by Gartner & Weiss (2013). We report the first demonstration that these traits predicted problem-solving success on a test of innovation. Two facets, Fearful/Aggressive and Cautious, significantly negatively predicted success in this task. This work should be considered preliminary, but we hope the promising results encourage future studies with larger sample sizes and further refinement of the behavioral traits measure. Felid behavioral trait research, in combination with cognitive testing, has practical applications for both captive welfare and wildlife conservation success.

Supplemental Information

Supplemental Information 1 Raw Data for Personality and MAB.

Click here for additional data file.

Supplemental Information 2 Codebook.

Click here for additional data file.

The authors wish to thank all who provided assessments and/or puzzle box testing assistance for this research: L. Barrett, P. Thomas, R. Snyder, T. Scarberry, R. Aversa, T. Teegan, T. Sinclair, E. White, R. Meo-Henry, L. Harney, B. Albert, N. Turner, C. Hood, S. Jamalapuram, A. Birk, P. Billette, K. Ellis, K. Wilson, S. Fantuzzi, A. Blanco, J. Kleoudis, V. Hussey, J. Neiss, B. O’Meara, E. Mowatt, C. Walsh, T. Gunther, S. Nelson, C. Norton, S. Spillman, R. Orens, T.L. Rossit, J. Jeffords, A. Cook, Kelly, M. Townten, R. Miyajima, L. Hayes, K. Templeton, K. Flatley, N. Borrego and R. Sides.

Additional Information and Declarations

Competing Interests

Author Contributions

Data Availability

Victoria O’Connor declares that she has no competing interests. Jennifer Vonk is a Section Editor for PeerJ.

Victoria L. O’Connor conceived and designed the experiments, performed the experiments, analyzed the data, prepared figures and/or tables, authored or reviewed drafts of the article, and approved the final draft.

Jennifer Vonk conceived and designed the experiments, analyzed the data, authored or reviewed drafts of the article, and approved the final draft.

The following information was supplied regarding data availability:

The raw data is available in the Supplemental Files.

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
