# Peer review of "Scaredy-cats don’t succeed: behavioral traits predict problem-solving success in captive felidae"

_PeerJ, doi:10.7717/peerj.14453_

## Round 0.1 · original submission · Major Revisions

All three referees provided valuable comments - I invite the Authors to adjust the manuscript accordingly. Please pay special attention to the validity of the statistical approaches used - two reviewers devoted some space in their reports to this aspect, especially to remove any subjectivity in forming and determining the PCs to use in analysis.

I also have some more general comments in addition to the referees comments regarding the analytical approach used.

(1) The data has clear hierarchical and clustered structure - I strongly recommend using the clustering (e.g., the identities of scorers) as a random factor in analysis. Without this the results are not robust and may represent falsely inflated type I error rates.

(2) As you analyses data from several species, there is strong possibility that resulting correlations may represent a Simpson paradox. To avoid the impact of phylogenetic interdependence you should analyse all your mixed models in a comparative framework, suing phylogenetic tree of all species as one of the random effects.

(3) Your paper lacks any figure that would help to understand the results. I strongly recommend including one for the linear model looking at solving success vs "personality" traits, and one visualising the PCA.

And finally one very important remark. Throughout the paper you refer to the traits analysed as "personality". This is not correct as in no case you actually have measured real personality (which is defined as consistent behaviour that is observed at multiple testing occasions and often along an environmental gradient). Personality can only be estimated in a mixed-model approach where individual behaviour is scarred repeatedly and individual identity is included in the model. In such framework analysis correlations e.g. with solving ability would be very easy as they could be directly estimated in the model through appropriate covariance structures. Also, it is not technically correct what you write in the first paragraph of the paper - namely, that personality = behavioural syndrome. Personality is a consistent within-individual variation in behaviour, and behavioural syndromes are within-individual correlation between different personality traits. Having said that - I do not think this invalidates your study. You have to however very careful with nomenclature and instead o using the "personality" term I would suggest just behaviour, behavioural type or something similar.

If you would be able to successfully address my and referees' comments I'm sure your paper could be a valuable contribution to the field of behavioural biology.

Reviewer 1 ·

Basic reporting

The Introduction should be completely rewritten to focus on the study at hand, and less of a treatise on personality, much of which has been published numerous times already.

The authors refer to subspecies as Pantherinae and Felinae--correct terminology is needed to avoid confusion.

Experimental design

There is no explanation for why the authors felt looking at the personality of multiple species as one might be valid, and they failed to mention any pitfalls with this approach. Does every species listed have a valid personality structure on its own?

A confound was mentioned for OKC but not addressed statistically or in any way other than mentioning it. This data should be removed from calculation.

Authors claim that reliability in personality testing is a requirement, yet then imply that they had one keeper doing assessments in some instances. This data should be removed from calculation.

Using eigenvalues from a PCA alone is generally not a good statistical method for determining factors. Parallel analysis should also be employed. Usually eigenvalues over 1 are used. Usually loading of over .4 are used. The authors claim "exceptions" were made in interpretation of the PCA results, which led to changing them based only on "better conceptual fit"--changing statistical results is not a valid methodology. Two "factors" had only one trait listed.

Reporting on reliability results needs to be more exact (using wording such as "majority" is not exact). Mean reliabilities are not reported.

Understanding statistics is important--"approaching significance" means nothing other than the data were not significant.

Validity of the findings

Based on the comments in the experimental design section, these results need to adjusted before I could assess validity.

Authors claim they examined whether personality predicts innovation, but there are other explanations here--for example, perseverance.

Reviewer 2 ·

Basic reporting

This is a well-written and clear paper assessing personality facets in Felidae and whether these facets can predict problem-solving performance on a well-established innovation paradigm, the multi-access puzzle box. Some of the strengths of the paper include relatively new animal models that have been tested. The cognition literature has been indeed heavily dominated by work on specific animal groups, and a more complete taxonomic distribution of studies on innovation will help provide important insight into the evolution of this ability. From my understanding, I see no fundamental flaws in the experimental design or interpretation of the results. As acknowledged by the authors, this study is preliminary and interpretations are made with parsimony throughout the manuscript. The principal uncorrectable weakness regarding this manuscript concerns the low sample size investigated (e.g., one individual of the Siberian lynx (Lynx lynx wrangeli)'s species has been investigated in the current study), which can cause the conclusions to lose robustness. However, this latter point is properly acknowledged by the authors in the Discussion section and I appreciate it may be difficult to have a larger sample with animals kept in captivity or with threatened animals in general.
I have some suggestions, which are listed below:
- Although the link between the first part of the title (before colon) and the proverb is well found and eye-catching, it seems that it does not truly reflect the results obtained. There were four personality facets that predicted success (or approached significance), and ‘Curiosity’ was not the best predictor among them. Perhaps the first part of the title may be revised and/or deleted?
- The keywords provided do not include personality, which may be more relevant than behavioral flexibility regarding the current study.
- Although background information and foundation of objectives have been well exposed in the introduction section, some parts may benefit from the addition of relevant and more recent references. For instance, although stated otherwise, in lines 42-54 references cannot be considered as ‘recent’ studies (e.g., Carere et al., 2010). Could you provide more recent references or remove this ‘recent’ statement from the text? The recent meta-analysis provided by Dougherty et al. (2018) may also be relevant to be cited in the text (Dougherty, L. R., & Guillette, L. M. (2018). Linking personality and cognition: a meta-analysis. Philosophical Transactions of the Royal Society B: Biological Sciences, 373(1756), 20170282).
- In lines 51-54: would it be possible to add a brief example along with the reference given in parentheses, for each point cited (zoo management, animal welfare, captive breeding, etc.)?
- In lines 102 & 103: you may remove the dashes and replace them by colons for clarity as facets already involve dashes.
- Lines 119-121: I find this sentence unclear, could you rephrase?
- In general, specifically because you used the multi-access puzzle box, it would be nice to provide definitions of innovation and behavioral flexibility within Introduction. It seems that sometimes those terms are used interchangeably within the text.
- Lines 135-136: to allow consistency throughout the text, replace slashes by dashes when speaking about dimensions (idem for the other sections e.g., Results, Discussion, and Conclusions).
- Line 142: ‘the current work’ is a bit confusing, perhaps modify it for instance by replacing it by ‘these latter studies’.
- Line 181: replace dash by colon for consistency.
- Line 183: what is meant by ‘neutral’? Perhaps give an example/short explanation in parentheses.
- Lines 187-189: is there a way to check whether knowledge of how individuals performed in the MAB (OKC subjects) represented a bias for keepers?
- Lines 193-194: a bit of an overstatement? Perhaps reformulate by replacing ‘a variety’ by ‘some’ given in O’Connor et al. (2022) two species were studied.
- In Results, as subjects belong to three different locations, it would be interesting to check whether problem-solving success differ within individuals.
- Lines 278-281: I would have made this statement in Introduction along with definitions of innovation and behavioral flexibility.
- Line 293: it seems there is an error in the sentence, remove the colon before curious.
- Line 299: could you provide a reference or short suggestions/examples to how questionnaires may be refined in future studies?
- Line 316: perhaps specify ‘with this group’.
- Line 320: perhaps refer to Table 2 in parentheses.
- Line 322: could you provide references?
- Line 323: are there more recent references in support of Mason and Latham (2004)?
- Lines 330-333: I would suggest to be more cautious with this claim - that the MAB (and problem-solving tasks in general that assess innovation) allows for examination of multiple measures of cognition - see for instance: van Horik, J. O., & Madden, J. R. (2016). A problem with problem solving: motivational traits, but not cognition, predict success on novel operant foraging tasks. Animal Behaviour, 114, 189-198.
- Lines 346-348: could you provide an example supporting this claim? For instance, how the knowledge acquired from future studies combining personality/cognition could help develop practical applications for your studied species or Felids more generally?
- Figures of the apparatus are a bit hard to understand at first glance. Perhaps you could divide Figure 1 and Figure 3 as you did for Figure 2 (with a white line separating left and right figure items) and then label and describe each figure a bit more, for instance by adding arrow(s) or specifying that the left figure shows the part of the apparatus consisting of which option.

Experimental design

The article describes interesting research that meets PeerJ’s Aims & Scope and ethical standards. Furthermore, the experiment clearly defines the research question, the knowledge gap being investigated, and seem to have been correctly designed and adequate for the questions pursued. However, I have some remarks, which are outlined below:
Some details are lacking if another researcher wants to replicate this experimental protocol properly. For instance, were the subjects food deprived? At what time of the day the trial was administered? Were they habituated to the human experimenter in any way? What was the exact procedure carried out by the experimenter during a trial? Has there been previous studies on these individuals? You could use the STRANGE framework to describe your population (Webster & Rutz, 2020, Nature, 582:337-340). Moreover, it is a little bit confusing what is meant by ‘Condition’ here, perhaps you may provide a very brief title specifying a bit more the content of each condition rather than ‘Condition 1’ and ‘Condition 2’?

Validity of the findings

Data are available and statistically sound, and the interpretations made by the authors are in accordance with the statistics carried out on the obtained results. The conclusions are cautious and acknowledge the pilot aspect of the task used in this study, and are well connected to the original question investigated.

Additional comments

I have no additional comments.

Reviewer 3 ·

Basic reporting

The authors should investigate the difference between personality traits and cognitive capacities deeply because they include one of the personality facets, intelligence, as a measure of their performance in learning and memory tasks, which is clearly not part of their personality but of their cognitive ability. Also, and related to this topic, the authors did not cite three fundamental works regarding the link between personality and cognition, i.e. Carere and Locurto 2011; Sih and Del Giudice 2012; Griffin et al. 2015, which most certainly would improve their concepts about this issue.

Experimental design

As I have mentioned before, "smart" or "intelligent" is not a personality trait. If the authors had a score of this "attribute" it should be analyzed separately from the personality PCA. Moreover, the authors should clarify whether the individuals tested were previously exposed to a test like that with the MAB.

Validity of the findings

the authors should analyze again the personality trait and run a model with the principal component and the scores from the intelligence facet as predicted variables of problem-solving success, and then discuss the results found. The validity of their findings depends on this.

Additional comments

Additional points to be corrected:
line 21. Eliminate the words "behavioural flexibility" because it is not a cognitive capacity, but a composite of cognitive and non-cognitive traits.
line 22. Intelligence is not a personality trait!
line 36. Only social decision-making?
line 37. The performance in cognitive tasks can or cannot be related to variation in personality, so, certainly, it is not a personality trait.
line 56. The authors must include a cite here.

---

## Round 0.2 · Minor Revisions

Dear Authors, as you can see below your paper was re-reviewed by 2 previous referees. They both acknowledged your corrections and are satisfied with the current version.

Before acceptance I would however suggest a few additional corrections that would make the paper clearer and more transparent. In the results section you mention the ICC as ICC(n,k) - this notation is not clear to me, so it would be good to explain what the numbers in parentheses mean (and, if this si some sort of multivariate analogue of pairwise ICC - to add a short explanation in methods). Similarly, you mention a parameter B in statistical tests, it is not clear from which analysis it comes from. Same for "Wald" - if this is a Wald test you should mention it in statistical methods, indicating in what type of procedure it was used. One last remark - the figure illustrating the factor analysis is unfortunately difficult to read, the points are "flat" and without any reference to the axes it is difficult to say how they cluster. My suggestion is to either add some additional perspective to the plot (intermediate grid lines and wall panels, point shading to make them look more 3d and dotted lines dropping from points to the bottom wall to indicate their positions in 3d more precisely) - or change the 3d plot to 3 pairwise 2d plots showing 3 possible combinations of axes.

Reviewer 1 ·

Basic reporting

The Basic reporting is much improved.

Experimental design

The authors have addressed all my original concerns here.

Validity of the findings

The authors have addressed my original concerns.

Reviewer 3 ·

Basic reporting

The revised version of this manuscript has improved considerably.

Experimental design

This area is clearer now.

Validity of the findings

no comments

Additional comments

As I mentioned above, the revised version of the manuscript has improved considerably. I don't have any additional comments or suggestions to give.

---

## Round 0.3 · accepted · Accept

The Authors have addressed all comments from the reviewers and myself, I'm happy to accept the manuscript for publication in its current form.